# Gull’s Theorem Revisited

**DOI:** 10.3390/e24050679

**Published:** 2022-05-11

**Authors:** Richard D. Gill

**Affiliations:** Mathematical Institute, Leiden University, P.O. Box 9512, 2300 RA Leiden, The Netherlands; gill@math.leidenuniv.nl

**Keywords:** Bell’s theorem, quantum entanglement, Monte Carlo simulation, singlet correlations, EPR-B model, local realism, quantum foundations

## Abstract

In 2016, Steve Gull has outlined has outlined a proof of Bell’s theorem using Fourier theory. Gull’s philosophy is that Bell’s theorem (or perhaps a key lemma in its proof) can be seen as a no-go theorem for a project in distributed computing with classical, not quantum, computers. We present his argument, correcting misprints and filling gaps. In his argument, there were two completely separated computers in the network. We need three in order to fill all the gaps in his proof: a third computer supplies a stream of random numbers to the two computers representing the two measurement stations in Bell’s work. One could also imagine that computer replaced by a cloned, virtual computer, generating the same pseudo-random numbers within each of Alice and Bob’s computers. Either way, we need an assumption of the presence of shared i.i.d. randomness in the form of a synchronised sequence of realisations of i.i.d. hidden variables underlying the otherwise deterministic physics of the sequence of trials. Gull’s proof then just needs a third step: rewriting an expectation as the expectation of a conditional expectation given the hidden variables.

## 1. Introduction

This paper is about a simple programming challenge—simple to state, that is. Can you set up two computers that will each separately receive a stream of angles and each output a stream of completely random plus or minus ones, such that the correlation between the outputs given the inputs is minus the cosine of the difference between the input angles? (See Figure 1). The answer is “no”. Many conditional correlation functions can be created since, after all, the two computers can have an identical pseudo-random generator built in, with the same seed and the same parameters, and hence, they can certainly create correlated pseudo-randomness. However, they cannot, conditional on the pair of settings, create the negative cosine of the difference between the angles. That statement is not obvious. It follows from Bell’s theorem and is rightly seen as a unique signature of quantum entanglement.

There are subtleties involved. What exactly is meant by “correlation”? Many experimenters proudly exhibited a negative cosine, but the amplitude of that curve is crucial. The question is: can classical computers generate a full amplitude negative cosine, where the “correlation” is the average of the product of the +/−1 valued outcomes. No post-selection or renormalisation is allowed.

We give a complete probabilistic proof of this impossibility theorem using basic results from Fourier analysis. No knowledge of quantum mechanics is needed.

The problem does come from a fundamental question in quantum physics, going back to the debates between Einstein and Bohr, the EPR paradox, and Bell’s (1964) theorem [1]. It is also very relevant to recent experimental progress such as the “Delft Bell experiment” of Hensen et al. (2015) reported in *Nature* [2]. A survey is given by the present author Gill (2014) [3].

Gull (2016) [4] outlined a novel connection to Fourier analysis in overhead sheets, which he used in several talks. Much of the text in the first subsection below is also part of the paper Gill (2021) [5]. In that paper, Gull’s theorem is of side interest and is mentioned but not proven. That led to the necessity to clarify the status of Gull’s proof. Gull and Bell were physicists, not in the business of writing out formal mathematical theorems. Bell himself referred to his *inequality* as his “theorem”, though, from a purely mathematical point of view, it is an elementary application of Boole’s fundamental inequality from the 19th-century prehistory of probability theory. On the other hand, the incompatibility of quantum mechanics and local realism can be framed as a formal mathematical theorem. Then the Bell inequalities are just a useful lemma in proving an important result in the foundations of physics. Not a deep result either, but a result with many metaphysical ramifications. See also Gill (2021) [6,7,8].

### Bell’s Theorem as a Theorem of Distributed Computing

We would like to paraphrase what is now called Bell’s theorem as the metaphysical statement that quantum mechanics (QM) is incompatible with local realism (LR). More precisely, and following B. Tsirelson’s wonderful Citizendium.org article [9], Bell’s theorem states that conventional quantum mechanics is a mathematical structure incompatible with the conjunction of three mathematical properties: *relativistic local causality* (commonly abbreviated to “locality”), *counterfactual definiteness* (“realism”) and *no-conspiracy* (“freedom”) (some readers have objected to various parts of this terminology; there is a continuously raging discussion in the philosophy of science concerning terminology). By conventional quantum mechanics, we mean: quantum mechanics including the Born rule, but with a minimum of further interpretational baggage. Whether the physicist likes to think of probabilities in a Bayesian or in a frequentist sense is up to them. In Many Worlds interpretations (and some other approaches), the Born rule is argued to follow from the deterministic (unitary evolution) part of the theory. Nevertheless, everyone agrees that it is there.

Bell himself sometimes used the phrase “my theorem” to refer to his *inequalities*: first his (Bell, 1964) three correlations inequality [1], and later what is now called the Bell-CHSH (Clauser, Horne, Shimony, Holt 1969) four correlations inequality [10]. We would rather see those inequalities as simple probabilistic *lemmas* used in two alternative, conventional, proofs of the same *theorem* (the incompatibility of QM and LR). Actually, the original proof involves four correlations, too, since it also builds on perfect anti-correlation in the singlet correlations when equal settings are used. Moreover, reference [1] also has a small section in which Bell shows that his conclusions still hold if all those correlations are only approximately true.

The CHSH inequality is, indeed, a very simple result in elementary probability theory. It is a direct consequence of the Boole inequality stating that the probability of the disjunction (the logical “or”) of several propositions is bounded by the sum of their individual probabilities. Something a whole lot stronger is Fine’s (1982) theorem [11], that the satisfaction of all eight one-sided CHSH inequalities together with the four no-signalling equalities is necessary and sufficient for a local hidden variables theory to explain the sixteen conditional probabilities p(x,y∣a,b) of pairs of binary outcomes *x*, *y*, given pairs of binary settings *a*, *b*, in a Bell-CHSH-type experiment. No-signalling is the statement that Alice cannot see from her statistics what Bob is doing: with typical statisticians’ and physicists’ abuse of notation (the abuse of notation is that which actual function you are talking about depends on the names you use for the variables on which it depends; this abuse is nowadays reinforced by being built into some computer languages), p(x∣a,b) does not depend on *b*, and similarly, for Bob, p(y∣a,b) does not depend on *a*.

The following remarks form a digression into the prehistory of Bell inequalities. With quite some imagination, a version of Fine’s theorem can be recognised in Boole’s (1853) monumental work [12], namely his book “*An investigation of the laws of thought, on which are founded the mathematical theories of logic and probabilities*”. See the end of Section 11 of Chapter XIX of [12], where the general method of that chapter is applied to Example 7, Case III of Chapter XVIII. Boole derives the conditions on three probabilities *p*, *q* and *r* of three events that must hold in order that a probability space exists on which those three events can be defined with precisely those three probabilities, given certain logical relations between those three events. In modern terminology let the events be A={X=Y}, B={Y=Z}, C={Z=X} where *X*, *Y* and *Z* are three random variables. It is impossible that two of the events are true but the third is false. Boole derives the six linear inequalities involving *p*, *q* and *r* whose simultaneous validity is necessary and sufficient for such a probability space to exist. Boole’s proof uses the arithmetic of Boolean logic, taking expectations of elementary linear identities concerning indicator functions (0/1 valued random variables), just as Bell did.

A similar exploration of necessary and sufficient conditions for a probability space to exist, supporting a collection of probability assignments to various composite events, was given by Vorob’ev (1982) [13]. Vorob’ev starts his impressive but nowadays pretty much forgotten paper with a little three variable example. He has no further concrete examples, just general theorems; no reference to Bell or to Boole. The fact that Bell’s inequalities have a precursor in Boole’s work was first mentioned (though without giving a precise reference) by Itamar Pitowsky in a number of publications in the early 1980s. This led several later authors to accuse Bell of carelessness and even to suggest plagiarism because Bell does not refer to Boole. At least one cannot blame Bell (1964) for not citing Vorob’ev’s paper. All that this really shows is that Bell’s theorem elicits very strong emotions, both positive and negative, and that lots of physicists and even mathematicians do not know much probability theory.

Having cleared up terminology and attribution, we continue towards the contribution of Gull to the story.

In a Bell-CHSH type experiment, we have two locations or labs, in which two experimenters, Alice and Bob, can each input a binary setting to a device, which then generates a binary output. The setting corresponds to an intended choice of an angle, but only two angles are considered in each wing of the experiment. This is repeated, say *N* times. We will talk about a *run* of *Ntrials*. The settings are externally generated, perhaps by tossing coins or performing some other auxiliary experiment. The *N* trials of Alice and Bob are somehow synchronised; exactly how does not matter for the purposes of this paper, but in real experiments, the synchronisation is taken care of using clocks, and the spatial-temporal arrangement of the two labs is such that there is no way a signal carrying Alice’s *n*th setting, sent just before it is inserted into her device, could reach Bob’s lab before his device has generated its *n*th outcome, even if it were transmitted at the speed of light.

In actual fact, these experiments involve measurements of the “spin” of “quantum spin-half particles” (electrons, for instance), or alternatively, measurements of the polarisation of photons in the plane opposite to their directions of travel. The two settings, both of Alice and Bob, correspond to what are intended to be two *directions* (spin) or *orientations* (polarisation), usually in the plane but conceivably in three-dimensional space. Polarisation does not only have an orientation—horizontal or vertical with respect to any direction in the plane—but also a degree of ellipticity (anything from circular to linear), and it can be clockwise or anti-clockwise. Talking about spin: in the so-called singlet state of two maximally entangled spin-half quantum systems, one can conceivably measure each subsystem in any 3D direction whatsoever, and the resulting pair of ±1-valued outcomes $\left(X_{a},Y_{b}\right)$ would have the “correlation” $EX_{a}Y_{b}=-a\times b$. Marginally, they would be completely random, $EX_{a}=EY_{b}=0$. The quantum physics set-up is often called an EPR-B experiment: the Einstein, Podolsky, Rosen (1935) [14] thought experiment, transferred to spin by Bohm and Aharonov (1957) [15].

When translated to the polarization example, this joint probability distribution of two binary variables is often called Malus’ law. We will, however, stick to the spin-half example and call it “the singlet correlations”. Moreover, we will restrict attention to spin measured in directions in the plane. The archetypical example (though itself only a thought experiment) of such an experiment would involve two Stern–Gerlach devices and is a basic example in many quantum physics texts. Present day experiments use completely different physical systems. Nowadays, anyone can buy time on a quantum computer “in the cloud” and do the experiment themselves on an imperfect two-qubit quantum computer. One can perhaps also look forward to a future quantum internet, connecting two one-qubit quantum computers.

Now, Bell was actually interested in what one nowadays calls (possibly “stochastic”) “local hidden variables theories” (LHV). According to such a theory, the statistics predicted by quantum mechanics, and observed in experiments, are merely the reflection of a more classical underlying theory of an essentially deterministic and local nature. There might be local randomness, i.e., by definition, randomness inside the internal structure of individual particles, completely independent of the local randomness elsewhere. Think of “pseudo-random” processes going on inside particles. Think of modelling the whole of nature as a huge stochastic cellular automaton. Mathematically, one might characterise such theories as claiming the mathematical existence of a classical probability space on which are defined a large collection of random variables $X_{a}$ and $Y_{b}$ for all directions *a* and *b* in the plane, such that each pair $\left(X_{a},Y_{b}\right)$ has the previously described joint probability distribution. Therefore, the question addressed by this paper is: can such a probability space exist? The answer is well-known to be “no”, and the usual proof is via the Bell-CHSH inequality. The impossibility theorem is what we will call *Gull’s theorem*. We are interested here in a different way to prove it—very different from the usual proofs—based on the outline presented by Gull.

The underlying probability space is usually called Λ instead of Ω, and the elementary outcomes λ∈Λ stand for the configuration of all the particles involved in the whole combined set-up of a source connected to two distant detectors, which are fed the settings *a* and *b* from outside. Thus $X_{a}\left(\lambda \right)$ stands *within the mathematical model* for the outcome that Alice would theoretically see if she used the setting *a*, even if she actually used another, or even if the whole system was destroyed before the outcomes of the actually chosen settings were registered. There is no claim that these variables exist in reality, whatever that means. I am talking about the *mathematical* existence of a mathematical model with certain *mathematical* features. I am not entering a debate involving use of the concepts *ontological* and *epistemological*. This is not about the properties of the real physical world. It is about mathematical descriptions thereof; mathematical descriptions that could be adequate tools for predictions of statistics (empirical averages) and predictions of probabilities (empirical relative frequencies). If I write about the outcome that Alice would have observed, had, counter to actual fact, Bob’s measurement setting been different from what he actually chose, I use colourful, and hopefully helpful, language about mathematical variables in mathematical models or about variables in computer code in computer simulation programs.

In a sequence of trials, one would suppose that for each trial there is some kind of resetting of the apparatus so that at the *n*th trial, we see the outcomes corresponding to $\lambda =\lambda_{n}$, where the sequence $\lambda_{1}$, $\lambda_{2}$, ..., represents independent draws from the same probability measure on the same probability space Λ. Now suppose we could come up with such a theory and indeed come up with a (classical) Monte-Carlo computer simulation of that theory on a classical PC. Then, we could do the following. Simulate *N* outcomes of the hidden variable λ, and simply write them into two computer programs as *N* constants defined in the preamble to the programs. More conveniently, if they were simulated by a pseudo random number generator (RNG), then we could write the constants used in the generator, and an initial seed, as just a few constants, and reproduce the RNG itself inside both programs. The programs are to be run on two computers thought of as belonging to Alice and Bob. The two programs are started. They both set up a dialogue (a loop). Initially, *n* is set to 1. Alice’s computer prints the message, “Alice, this is trial number n=…. Please input an angle.” Alice’s computer then waits for Alice to type an angle and hit the “enter” key. Bob’s computer does exactly the same thing, repeatedly asking Bob for an angle.

If, on her *n*th trial, Alice submits the angle *a*, then the program on her computer evaluates and outputs $X_{a}\left(\lambda_{n}\right)=\pm 1$, increments *n* by one, and the dialogue is repeated. Alice’s computer doesn’t need Bob’s angle for this – this is where locality comes in! Bob’s does not need Alice’s.

We have argued that if we could implement a local hidden variables theory in *one* computer program, then we could simulate the singlet correlations derived from one run of many trials on *two completely separate* computers, each running its own program, and each receiving its own stream of inputs (settings) and generating its own stream of outputs.

To summarise: for us, a local hidden variables theory for the EPR-B “Gedanken experiment”, including what is often called a *stochastic* local hidden variables theory, is just a pair of functions A(a,λ), B(b,λ) taking values ±1, where *a* and *b* are directions in the plane, together with a probability distribution over the third variable λ, which can lie in any space of any complexity and which represents everything that determines the final measured outcomes throughout the whole combined system of a source, transmission lines, and two measurement stations, including (local, pseudo-) randomness there. In the theory, the outputs are a deterministic function of the inputs. When implemented as two computer programs, even more is true: *given the programs*, the *n*th output of either computer depends *only* on *n* and on the *n*th input setting angle on that computer, not on the earlier input angles. If one reruns either program with an identical input stream, the output stream will be the same, too. In order to obtain different outputs, one would have to change the constants defined in the program. As we mentioned before, but want to emphasise again, we envisage that an identical stream of instances of the hidden variable λ is generated on both computers by the same RNG, initialised by the same seed; that seed is a constant fixed in the start of the programs.

The previous discussion was quite lengthy but is needed to clear up questions about Gull’s assumptions. Actually, quite a lot of careful thought lies behind them.

Gull [4] posed the problem: write those computer programs, or prove that they do not exist. He gave a sketch of a rather pretty proof that such programs could not exist using Fourier theory, and that is what we will turn to next. We will call the statement that such programs do not exist *Gull’s theorem*.

We will go through Gull’s outline proof but will run into difficulty at the last step. However, it can easily be fixed. In Gull’s argument, there are, after some preparations, two completely separated computers running completely deterministically. We need three networked computers: a third computer supplies a stream of random numbers to the two computers, which represent the two measurement stations in Bell’s theorem. At the end of the day, one can imagine that computer replaced by a cloned, virtual computer, generating the same pseudo-random numbers within each of Alice and Bob’s computers. Gull’s proof then just needs a third step: writing a grand expectation over all randomness as the expectation of a conditional expectation, given the hidden variables.

A recently posted stackexchange discussion [16]) also attempts to decode Gull’s proof, but in our opinion, is also incomplete, becoming stuck at the same point as we did.

In a final section, we will show how *Gull’s theorem* (with completely separated, deterministically operating computers) can also be proven using a Bell theorem proof variant due to the first author of the present paper, designed specifically for fighting Bell deniers by challenging them to implement their theory as a networked computer experiment. The trick is to use externally created streams of random binary setting choices and derive martingale properties of a suitable game score, treating the physics implemented inside the computers as completely deterministic. Randomness resides only in the streams of settings used, trial by trial, in the experiment.

The message of this paper is that *Gull’s theorem* is true. Its mathematical formulation is open to several interpretations, needing different proofs, of course. A version certainly can be proven using Fourier theory, and Gull’s Fourier theoretic proof of this version is very pretty and original indeed.

We have referred to the question of computer simulation of Bell experiments, and to Gull’s proof of Bell’s theorem, in recent papers by Gill (2021, 2022). It is good that any doubts as to the validity of Gull’s claim can be dispelled, though his claim does need more precise formulation since his outline proof has a gap. We hope he would approve of how we have bridged the gap.

## 2. Gull’s Theorem: A Distributed Computer Simulation of the Singlet Correlations Is Impossible

The reader needs to recall the standard theory of Fourier series, whereby one approximates a square-integrable complex-valued function on the circle by an infinite sum of complex Fourier coefficients times periodic exponential functions. A useful introductory resource (freely available on the internet) is the book by Schoenstadt (1992) [17]. In particular, one needs Section 2.6, The Complex Form of the Fourier Series; Section 6.2, Convolution and Fourier Transforms; and Section 6.9, Correlation. However, the book does not go into measure theoretic niceties. At the very least, an assumption of measurability needs to be made in order that integrals are well defined (or instead of Lebesgue integrals, one has to use the notion of Radon integrals). The Fourier series transform converts a square-integrable complex-valued function to a square summable doubly infinite sequence of coefficients. With appropriate normalisations, one has an isomorphism between the Hilbert spaces $L_{C}^{2}\left(-\pi ,\pi \right)$ and $\ell_{C}^{2}\left(Z\right)$. Said in another way, the periodic complex exponential functions on the circle, appropriately normalised, form a complete orthonormal basis of the function space of equivalence classes of all square-integrable functions on the circle.

The transformation *in each direction* is called by Gull a Fourier transform, and by the theory of *Pontryagin duality,* this is reasonable terminology. The fact that a convolution of functions maps to a product of a Fourier series is very well known. Less well-known is that a similar result holds for the correlation between two functions.

The analysis literature has numerous results giving stronger forms of convergence of the Fourier series to the approximated function than the $L^{2}$ convergence coming from the point of view just explained. One could avoid analytic niceties by working with discrete settings, for instance, whole numbers of degrees, and using the discrete Fourier transform. Of course, when one selects a whole number of degrees on a machine with a digital interface, what goes on inside the machine will likely involve angles, which are never precisely the angles the user asked for. However, that does not matter: the experimenter sets a digital dial to some whole number of degrees. One could say that this is a Bell-CHSH experiment in which Alice and Bob each have 360 different settings at their disposal.

We will not develop that approach in this short paper but do plan to explore it in the near future. We will just present Gull’s argument as it stands and see where the argument needs fixing.

According to the singlet correlations, if Alice and Bob submit the same sequence of inputs, then their output streams are exactly opposite to one another. Recall that the outcomes are functions only of the trial number and the respective setting angles. Let us denote them by $A_{n}\left(\theta \right)$ and $B_{n}\left(\phi \right)$. Then, we must have the following equality between functions, defined, say, on the interval (−π,π]: $B_{n}=-A_{n}$.

Next, consider a sequence of *N* trials in which Alice uses angles, picked uniformly at random from that interval, and Bob uses the same sequence of angles, shifted clockwise by the amount θ. The expectation value of the average of the products of their outcomes is
(1)$$\rho_{N}\left(\theta \right)\;=\;-\frac{1}{N}\;\sum_{n=1}^{N}\;\;\frac{1}{2\pi }\;\;\;\int_{u\in (-\pi ,\pi ]}A_{n}\left(u\right)A_{n}\left(u-\theta \right)\;du.$$

We want this “expected sample correlation function” $\rho_{N}$ to converge (pointwise) to the negative cosine, the correlation function ρ defined by
(2)$$\rho \left(\theta \right)\;:=\;-\cos\left(\theta \right)\;=\;-\frac{1}{2}e^{i\theta }-\frac{1}{2}e^{-i\theta }$$
as *N* converges to infinity.

Notice, we already took the expectation value with respect to the randomness of the sequence of setting pairs. We actually want convergence almost surely: in a single infinitely long experiment, we already want to see the negative cosine law. For this, we could use the strong law of large numbers in the case of non-identically distributed but bounded random variables. That would tell us that if $\rho_{N}$ converges to a limit, then the “measured correlation” will converge also, with probability one.

Gull does not enter into any of these subtleties. Moreover, at some point, he drops the index *n* from his notation.

Suppose $A_{n}$ has Fourier series $\left({\tilde{A}}_{n}\left(k\right):k\in Z\right)$, i.e., with
(3)$${\tilde{A}}_{n}\left(k\right)\;=\;\frac{1}{2\pi }\;\;\;\int_{u\in (-\pi ,\pi ]}e^{-iku}A_{n}\left(u\right)\;du,$$
(4)$$A_{n}\left(u\right)\;=\;\sum_{k\in Z}{\tilde{A}}_{n}\left(k\right)e^{iku}.$$

This requires some regularity of each function $A_{n}$. However, assuming (Equation 4), we can substitute it into (Equation 1), twice, giving us a triple summation over *n*, *k*, and $k^{\prime }$ (say), with coefficient ${\tilde{A}}_{n}\left(k\right){\tilde{A}}_{n}\left(k^{\prime }\right)$, and a single integration over *u* of $e^{-iku}e^{-ik^{\prime }\left(u-\theta \right)}/2\pi$. The integral over the circle of $e^{-(k+k^{\prime })u}/2\pi$ is zero unless $k+k^{\prime }=0$, when it equals 1. We obtain the summation over *n* and *k* of ${\tilde{A}}_{n}\left(k\right){\tilde{A}}_{n}\left(-k\right)e^{-ik\theta }$. From (Equation 3), because the function $A_{n}$ is real, we obtain
(5)$$\rho_{N}\left(\theta \right)\;=\;-\frac{1}{N}\sum_{n=1}^{N}\sum_{k\in Z}{\left|{\tilde{A}}_{n}\left(k\right)\right|}^{2}e^{-ik\theta }.$$

Now, we already saw the Fourier series for the negative cosine: it just has two non-zero coefficients, obviously; those for k=±1. This tells us that for all k≠±1, the limit as *N* tends to infinity of $\sum_{n=1}^{N}{\left|{\tilde{A}}_{n}\left(k\right)\right|}^{2}/N$ is zero. We know that the real functions $A_{n}$ square to +1. However, only the k=±1 terms in their Fourier expansions play asymptotically a significant role in their expansions. However, on average, and in $L^{2}$ sense, they approach the negative cosine. Intuitively, this must be impossible. However, we do not presently have a proof.

The problem is all those different functions $A_{n}$. Gull, in his notes, is clearly thinking of there just being one.

The solution we propose is to introduce a third computer; one could consider it to represent the source. It supplies a stream of uniform random numbers to computers A and B, and now, inside the dialogue loop, the *same* function is called at each iteration, whose inputs consist *only* of the *n*th angle and the *n*th random number. Denote a generic random number by λ∈Λ with probability distribution P. Fix θ and choose, uniformly at random, *N* setting pairs, an angle θ apart. We can use Equation (Equation 1) again in which, by definition now,
(6)$$A_{n}\left(u\right)=A\left(u,\lambda_{n}\right).$$

Equation (Equation 1) now represents the sample average of *N* independent, identically distributed random variables. We have taken the conditional expectation, given the hidden variables $\lambda_{1}$, ..., $\lambda_{n}$ of the correlation observed in a run of length *N*; we have averaged only over the setting choices. The expectation value of this, thus averaging both over hidden variables and over setting pairs at a fixed distance θ apart, then becomes
(7)$$\frac{1}{2\pi }\int_{\lambda \in \Lambda }\;\int_{u\in (-\pi ,\pi ]}A\left(u,\lambda \right)A\left(u-\theta ,\lambda \right)\;du\;dP\left(\lambda \right).$$

Using (Equation 6), Equation (Equation 4) can be rewritten as
(8)$$A\left(u\right)\;=\;\sum_{k\in Z}\tilde{A}\left(k\right)e^{iku}$$
where A(u) and $\tilde{A}\left(k\right)$ are both random variables. Substitute (Equation 8) into (Equation 7), twice, giving us again summations over *k*, and $k^{\prime }$ (say), with coefficient $\tilde{A}\left(k,\lambda \right)\tilde{A}\left(k^{\prime },\lambda \right)$ and integrals over *u* of $e^{-iku}e^{-ik^{\prime }\left(u-\theta \right)}/2\pi$, and over λ with respect to the probability measure P.

Now, the integral of $e^{-(k+k^{\prime })u}$ is zero unless $k+k^{\prime }=0$. Therefore we obtain, analogously to (Equation 5),
(9)$$\rho \left(\theta \right)\;=\;E\;\rho_{N}\left(\theta \right)\;=\;\sum_{k}E(|\tilde{A}\left(k\right){|}^{2})e^{-ik\theta }.$$

From (2), $E(|\tilde{A}\left(k\right){|}^{2})=0$ for all *k* except k=±1. The random measurement functions only have two non-zero Fourier series coefficients and cannot possibly be discontinuous, which they must be since they only take the values ±1.

A similar project was independently embarked on by H. Razmi (2005, 2007) [18,19]. His work was criticised by R. Tumulka (2009) [20], but Razmi (2009) [21] fought back. Razmi makes a further physical assumption of invariance under rotations. Gill (2020) [22] discusses how some invariances may be imposed “for free” since, at the end of the day, we are going to generate correlations that are invariant under the same rotation of both settings.

### A Classical Distributed Computer Simulation of the Singlet Correlations Is Impossible

Quite a few years before the work of Gull, Razmi and Tumulka, and deliberately focussed on classical computer simulations of CHSH-style experiments. In 2003, Gill [23,24] produced several martingale-based inequalities on a variant of the CHSH statistic. He noticed that if the denominators of the four numerators in the four sample correlations were replaced by their expectation values (when settings are chosen completely at random) N/4, and then the whole statistic is multiplied by N/4, then it equals a sum over the *N* trials of a quantity which, under local realism, and assuming the complete randomness of the *setting* choices only, has an expectation value of less than 3/4. Therefore, subtract off (3/4)N and one has, under local realism, a supermartingale. The conditional independence of increments of the process (the time variable being *N*), conditional on the past, are negative. Moreover the increments of the process are bounded, so powerful martingale inequalities give easy exponential inequalities on the probabilities of large deviations upwards.

These inequalities were later improved and the martingale structure further exploited to obtain wonderfully sharp results. In our opinion, the nicest is that presented in the supplementary material of the paper Hensen et al. (2015) published in *Nature* [2], as the first ever successful “loophole-free” Bell type experiment. Since those researchers were based in Delft, the Netherlands, and have had much contact with the authors, we (the author) are particularly proud of that experiment and that probabilistic result.

Consider such an experiment and let us say that the *n*th trial results in a success, if and only if the two outcomes are equal and the settings are *not* both the setting with label “2”, or the two outcomes are opposite and the settings *are* both the setting with label “2”. (Recall that measurement outcomes take the values ±1; the settings will be labelled “1” and “2”, and these labels correspond to certain choices of measurement directions in each of the two wings of the experiment). The quantum engineering is set up so as to ensure a large positive correlation between the outcomes for setting pairs 11, 12 and 21, but a large negative correlation for setting pair 22. Let us denote the total number of successes in a fixed number, *N*, of trials, by $S_{N}$.

Then Hensen et al. (2015), and see also Bierhorst (2015) [25] and Elkouss and Wehner (2016) [26] for further generalisations, show that, for all *x*,
(10)$$P\left(S_{N}\geq x\right)\;\leq \;P\left(\mathrm{Bin}\left(N,3/4\right)\geq x\right),$$
where Bin(N,p) denotes a binomially distributed random variable with parameters *N*, the number of trials, and success probability, per independent trial, *p*.

Above, we wrote, “under the assumption of local realism”. Those are physics concepts. The important point here is that a network of two classical PCs both performing a completely deterministic computation, and allowed to communicate over a classical wired connection *between* every trial and the next, does satisfy those assumptions. The theorem applies to a classical distributed computer simulation of the usual quantum optics lab experiment. Time trends and time jumps in the simulated physics, and correlations (dependency) due to the use of memory of past settings (even of the past settings in the other wing in the experiment) do not destroy the theorem. It is driven *solely* by the random choice anew, trial after trial, of one of the four pairs of settings, and such that each computer is only fed its own setting, not that given to the other computer.

Take for instance N=10,000. Take a critical level of x=0.8N. Local realism says that $S_{N}$ is stochastically smaller (in the right tail) than the Bin(N,0.75) distribution. According to quantum mechanics, and using the optimal pairs of settings and the optimal quantum state, $S_{N}$ has approximately the Bin(N,0.85) distribution. Under those two distributions, the probabilities of outcomes, respectively, larger and smaller than 0.80N are about $10^{-30}$ and $10^{-40}$, respectively. These probabilities give an excellent basis for making bets.

Suppose we use the computer programs in this way just to compute the four correlations between the pairs of settings used in the CHSH inequality. This comes down to looking at the correlation curve in just four points. If the programs yield correlations that converge to some limits as *N* tends to infinity, then (4) shows that those four limits will violate the CHSH inequality and, hence, cannot be the four points on the correlation function that the program is supposed to be able to reproduce, up to statistical variation in its Monte Carlo simulation results.

## 3. Conclusions

Gull’s Fourier theory proof of Bell’s theorem as a no-go theorem in distributed computing can be made rigorous after notational ambiguity is resolved and an extra step is introduced. To follow his proof sketch, we must first of all introduce randomness into the setting choices. In the main part of the proof, we must think of picking, uniformly at random, pairs of settings a fixed distance apart on the circle. Secondly, we must see the streams of simulated hidden variables used in any long run by both computers as coordinatewise functions of the realisations of a single stream of i.i.d. variables simulated by a pseudo-random generator, thus with the same seed and the same parameters on each computer. The computers must contain an implementation of a classical local hidden variables model, with two measurement functions A(a,λ) and B(b,λ) and one probability distribution of the hidden variable λ. There is no use of memory or time.

A proof via Fourier analysis requires regularity assumptions. This can certainly be avoided by using discrete settings (e.g., whole numbers of degrees) and discrete Fourier transforms. The restrictions that Gull implicitly makes can be avoided by using a proof based on the martingale ideas first used by Gill (2003) [23,24], which exploit the logic behind the CHSH inequality, though seeing randomness only in the setting choices, leaving the physics as something completely deterministic.

Marek Żukowski has introduced a concept of “functional Bell inequalities”, see especially [27]. Earlier work by many authors has shown that the largest amplitude of a negative cosine that can be observed in a typical Bell-type experiment, in which settings of Alice and Bob are taken from the whole unit circle, is $k=1/\sqrt{2}=0.7071\ldots$; in other words, the mean of the product of the outcomes ±1 at given settings can be −kcos(α−β) with k=0.7071…, but not with any larger value of *k*. This is, in fact, an example of the generalised Grothendieck inequality for dimension two. In higher dimensions (for instance, directions in space, rather than in the plane), the Grothendieck constants are a matter of conjecture and of numerical bounds; see [28]. A very interesting question is whether functional Bell test experiments actually have advantages over the usual experiments, in which Alice and Bob each choose between two possible settings, so only four different correlations are experimentally probed. It could well be that there is actually a statistical advantage in probing more pairs of directions. Reference [29] showed that when we measure the power of a proof of Bell’s theorem in terms of the experimental resources needed to obtain a given degree of experimental proof that local realism is not true, it is not necessarily true that the best experimental demonstration of quantum nonlocality is attained through the experiment corresponding most closely to the most simple or striking mathematical proof. In particular, proof of Bell’s theorem “without inequalities” typically does need inequalities before it can apply to an experimental set-up, and moreover, the number of resources needed can be surprisingly large.

## Figures and Tables

**Figure 1 entropy-24-00679-f001:**
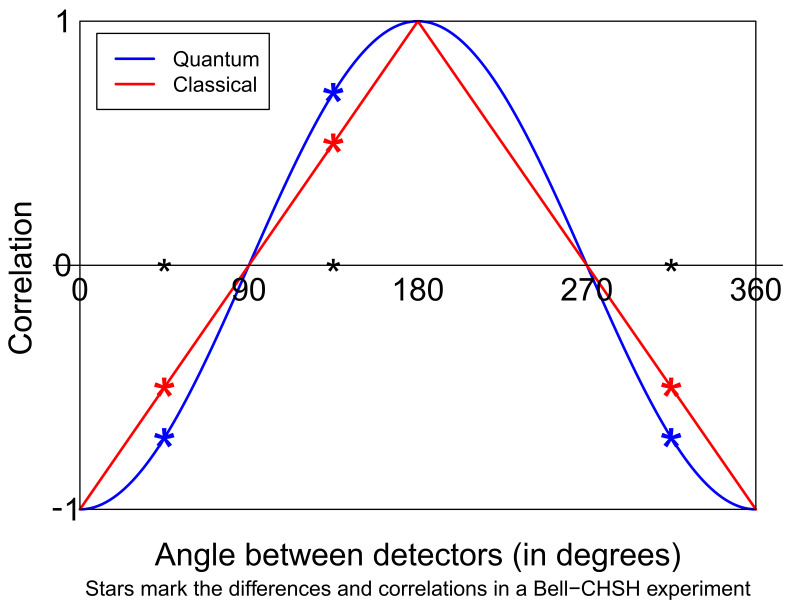
A possible correlation function allowed by quantum mechanics (blue) and one allowed by local realism (red). The asterisks mark the predictions for the angle pairs typically chosen in a Bell-type experiment. Alice uses angles $0^{\circ }$, $90^{\circ }$; Bob uses $45^{\circ }$ and $135^{\circ }$. The four differences (Bob minus Alice) are 45−0=45, 45−90=−45, 135−0=135, 135−90=45. One correlation is large and positive, three are large and negative.

## Data Availability

Not applicable.

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
