# Peer review of "Gull’s Theorem Revisited"

_entropy, 2022, doi:10.3390/e24050679_

Round 1

Reviewer 1 Report

The manuscript "Gull's theorem revisited" presents and completes a result by Steve Gull. While the original material is only available in the form of notes, the current presentation provides a good deal of context, both from the perspective of Bell non-locality and of Fourier analysis, which are the two domains of interest here. Whereas a textbook on Fourier analysis is cited, one can regret maybe that no reviews on Bell non-locality are cited.

Bell's result plays a significant role in modern science. The new derivation proposed by Gull and now Gill, called here 'Gull's theorem' is a nice addition to the literature. Its interest is to rely on a completely different derivation than used previously.

The current contribution is not restricted to presenting Gull's argument, but also improves it by filling a gap, thus turning it really into what can be called a theorem. The problem identified here is solved elegantly by noticing that the hidden variables for all rounds of a Bell test can be assumed to be provided just once, at the beginning of the whole experiment.

Section 2.1 raises a question: it is argued there basically that the p-value of a Bell test can be computed thanks to some recent results from 2015 and 2016, and that it is just given by the binomial tail. Since the random variables at hand are simply Bernoulli variables with unknown but bounded expectation value (even when conditioned on their past), isn't this a direct consequence of Thm. 4 of [W. Hoeffding, Ann. Math. Statist. 27 (3) 713 - 721, September, 1956]? None of the results mentioned here seem to be aware of this standard statistics result by Hoeffding...

Minor comments:

  • The introduction casts Bell's result as an elementary, essentially trivial result. While it is true that Bell's inequality can be derived from Boole's conditions of possible experience, and that this was not immediately noticed, it is now unequivocally recognized. Moreover, every result is trivial in some sense once proven, so it is not clear in which way such comments are helpful to the discussion. A more nuanced vocabulary could be used in case it it is not the intent of the author to despise Bell's work.
  • There are instances of 'the the' and 'ciricle'
  • The first sentence of Sec. 2.1 seems to be missing something

Author Response

I thank the referee for the careful reading. I have corrected the typos, and added a reference to a survey of Bell's theorem, in response to one of your suggestions. I also removed the disparaging word "trivial" concerning a step in Bell's proof. Concerning the p-values question: you are right that one can get *bounds* to p-values by using repeated conditioning and Hoeffding's inequality. That is in fact exactly what I did myself in two papers in 2001 and 2003. The fact that one can also get a *sharp bound* by repeated conditioning and construction of a coupled binomial random variable was a new result found by the Delft team in 2015, who were inspired by my earlier work, and cited it. This was already written in the paper, I think clearly enough.

Reviewer 2 Report

In this article, the author introduces Gull' theorem, which claim that the probability distribution produced from Bell state (maximally entangled state) cannot be obtained from independent classical random generators. Although the meaning of the theorem is equivalent to Bell's inequality, a bit different perspective of the theorem for hidden variables theories is beneficial for deeper understanding of quantum mechanical non-locality. I will recommend the paper to accept in Entropy in the present form.

Author Response

Thanks for your positive evaluation!

Reviewer 3 Report

As far as I understand, the main content of this paper is proposing an improvement for an earlier no-go theorem by Steve Gull, which states that distributed computation by classical computation can not simulate the measurement correlations of a singlet state by using Fourier analysis.
In the first section of the paper, the author makes a detailed introduction to the background of Bell's theorem, in which the emphasis is put on its mathematical foundation. Also, the author introduces a distributed computing task which is equivalent to a local hidden variable model, and performable with classical computers.
Then in the second section, the author introduces Gull's theorem. The original version of the theorem is just a few slides, in which some notations are only ambiguously defined, and a rigorous proof is not given. The author gives clear definitions to all the notations involved with the original theorem with his own interpretation, and shows that it is difficult to prove it directly because of the lack of i.i.d assumption for the local strategy. Then the author proposes a solution by introducing a third computer, which provides i.i.d random numbers to the two distributed computers. After the modification, the author proves rigorously with Fourier analysis that it is impossible to realize the singlet correlation by such a computing task.
I agree with the author's interpretation of the original Gull's theorem, and that the original proof needs to be fixed. The modification proposed in the paper is reasonable, and the relevant mathematics seem correct to me. I think applying Fourier analysis provides a simple and different way of proving Bell's theorem, and could be inspiring for some other relevant problems. This paper provides the first formal proof of the problem, together with many useful discussions. It is also well written and easy to read. So I would recommend acceptance of the paper.  

Author Response

Thanks for your positive evaluation!